Evaluation of potential reference genes for real-time qPCR analysis in a biparental beetle, Lethrus apterus (Coleoptera: Geotrupidae)

Nagy Nikoletta A. nnolett@gmail.com 1 2
Németh Zoltán 1 2
Juhász Edit 1
Póliska Szilárd 3
Rácz Rita 1 2
Kosztolányi András 2 4
Barta Zoltán 1 2
1 Department of Evolutionary Zoology, University of Debrecen , Debrecen , Hungary
2 MTA-DE Behavioural Ecology Research Group, University of Debrecen , Debrecen , Hungary
3 Genomic Medicine and Bioinformatic Core Facility, Department of Biochemistry and Molecular Biology, Faculty of Medicine, University of Debrecen , Debrecen , Hungary
4 Department of Ecology, University of Veterinary Medicine Budapest , Budapest , Hungary
Negri Ilaria
Electronic publication date: 2017 Nov 28
Publication date: 2017
Volume: 5
Electronic Location ID: e4047
Received 2017 Jul 29; Accepted 2017 Oct 26
Copyright: ©2017 Nagy et al.
Copyright year: 2017
Copyright holder: Nagy et al.
License: This is an open access article distributed under the terms of the Creative Commons Attribution License, which permits unrestricted use, distribution, reproduction and adaptation in any medium and for any purpose provided that it is properly attributed. For attribution, the original author(s), title, publication source (PeerJ) and either DOI or URL of the article must be cited.
License URL: https://creativecommons.org/licenses/by/4.0/

Keywords: Insect, Parental care, Housekeeping gene

Funding: NKFIH K112670 K112527 PD 121013 Hungarian Academy of Sciences European Union and the European Social Fund EFOP-3.6.1-16-2016-00022 The study was financed by NKFIH Grants (No. K112670 and No. K112527), and partially supported by the European Union and the European Social Fund through project EFOP-3.6.1-16-2016-00022, András Kosztolányi was supported by the János Bolyai Research Scholarship of the Hungarian Academy of Sciences and Zoltán Németh was supported by an NKFIH Fellowship (No. PD121013). There was no additional external funding received for this study. The funders had no role in study design, data collection and analysis, decision to publish, or preparation of the manuscript.

==============================
Hormones play an important role in the regulation of physiological, developmental and behavioural processes. Many of these mechanisms in insects, however, are still not well understood. One way to investigate hormonal regulation is to analyse gene expression patterns of hormones and their receptors by real-time quantitative polymerase chain reaction (RT-qPCR). This method, however, requires stably expressed reference genes for normalisation. In the present study, we evaluated 11 candidate housekeeping genes as reference genes in samples of Lethrus apterus, an earth-boring beetle with biparental care, collected from a natural population. For identifying the most stable genes we used the following computational methods: geNorm, NormFinder, BestKeeper, comparative delta Ct method and RefFinder. Based on our results, the two body regions sampled (head and thorax) differ in which genes are most stably expressed. We identified two candidate reference genes for each region investigated: ribosomal protein L7A and RP18 in samples extracted from the head, and ribosomal protein L7A and RP4 extracted from the muscles of the thorax. Additionally, L7A and RP18 appear to be the best reference genes for normalisation in all samples irrespective of body region. These reference genes can be used to study the hormonal regulation of reproduction and parental care in Lethrus apterus in the future.

Introduction

Hormonal regulation in insects generates great interest among entomologists but hormones have only been studied in detail in a few species (Gullan & Cranston, 2014). Insect hormones of particular interest include juvenile hormones, ecdysteroids and neuropeptides. These molecules regulate a vast number of physiological and developmental processes as well as behaviours (Gäde, Hoffmann & Spring, 1997). Studying these hormones used to be difficult considering their small amount and the occasional instability (Gullan & Cranston, 2014). The technical revolution of molecular biology and genetics, however, made it attainable to discover the details of genetic and hormonal regulation in insects (Raikhel, Brown & Belles, 2005). Some of the processes controlled by hormones mentioned above, such as ecdysis (Mykles et al., 2013), are already well described. Nevertheless, there are many other interesting physiological and behavioural mechanisms, like parental care, the hormonal regulation of which are not well understood (Panaitof et al., 2016). One way to increase our understanding of hormonal regulation is to identify patterns of gene expression associated with the hormones in question (Champagne & Curley, 2012).

Real-time quantitative polymerase chain reaction (RT-qPCR) is a commonly used method for analysing gene expression as it is a sensitive, fast and reproducible method; moreover, it requires only a minimal amount of RNA (Radonić et al., 2004). With this method, gene expression levels can be measured simultaneously in several different samples for a limited number of genes. Gene expression analyses with RT-qPCR, however, require some kind of normalisation in order to control the variation caused by stochastic processes occurring during the analytic procedure (Vandesompele et al., 2002). This normalisation is usually achieved by taking into account the expression level of so-called reference genes (VanGuilder, Vrana & Freeman, 2008). These genes are usually selected from housekeeping genes, which produce proteins vital for maintaining fundamental cell functions, like ribosomal or cytoskeletal proteins. Therefore, the expression levels of these reference genes are thought to be relatively stable. Thus, comparing the expression level of genes of interest with the expression levels of the reference genes, we can eliminate the differences caused by the different amount and quality of starting material. With this method we are able to control for differences occurring due to technical errors during sample preparation as well (e.g., RNA isolation and cDNA synthesis, Radonić et al., 2004). Nonetheless, expression levels of the housekeeping genes may also vary considerably under certain circumstances because they can be involved in processes other than maintenance functions of the cell, e.g., apoptosis (Nicholls, Li & Liu, 2012), cytokinesis (D’Souza-Schorey & Chavrier, 2006) and development (Zhou et al., 2015). Therefore, a given housekeeping gene cannot automatically serve as reference gene, and normalisation with unstable reference genes can lead to erroneous quantification results and conclusions (Thellin et al., 1999). Consequently, reference genes must be carefully selected so that their expression levels are similar between the different samples and should not be influenced significantly by different experimental conditions (VanGuilder, Vrana & Freeman, 2008). According to Vandesompele et al. (2002), the combination of two or more reference genes is highly recommended for normalisation to obtain more accurate results. In case of multiple reference genes, it is advised to use the geometric mean for normalisation since it better controls for extreme values and the possible differences between expression levels of the different genes (Vandesompele et al., 2002).

Lethrus apterus (Laxmann, 1770) (Coleoptera: Geotrupidae) is an earth-boring beetle that has biparental care during which the parents provision food for their offspring in advance their hatching (Kosztolányi et al., 2015). This kind of parental care is a complex and relatively rare trait among insects (Smiseth, Kölliker & Royle, 2012) and makes this beetle an outstanding model species for studying the hormonal background of parental care. In order to do so, however, stably expressed reference genes have to be identified.

In recent years, numerous studies aimed to identify stable reference genes in insects (Lord et al., 2010; Ponton et al., 2011; Bansal et al., 2012; Li et al., 2013; Pan et al., 2015; Yu et al., 2016). Nevertheless, there is a lack of reference gene studies that use individuals from natural populations. Our objective in this study was to examine the expression stability of several housekeeping genes in Lethrus apterus across different times of the breeding period in a natural population in order to identify the most stable reference gene(s). With the right combination of reference genes, the expression levels of hormone regulating genes involved in parental care in Lethrus apterus can be examined accurately in the future. Based on the literature (Shi et al., 2013; Liang et al., 2014; Zhu et al., 2014; Yang et al., 2015), we probed eleven housekeeping genes.

Materials and Methods

Sample collection

Samples were collected near Dorogháza, northern Hungary (47°59′29″N, 19°53′36″E) on 16th April, 4th May and 28th May in 2015, which dates corresponded to the beginning, middle and end of the breeding season of Lethrus apterus, respectively. Sample collection was approved by the Nothern Hungarian Inspectorate for Environment Protection and Nature Conservation (No. 9007-8/2014). The first sampling date represents the period of mate choice, while the second and third samplings were done during the period when parents were collecting leaves for the offspring. On each sampling dates, head and thorax samples were collected from eight males and eight females. All tissues were removed from the head capsule and muscle samples were taken from the thorax. Samples were collected in the field in less than five minutes after euthanizing the individuals. Each head and thorax sample was put immediately into separate eppendorf tubes which already contained 600 µl RNAlater® Stabilization Solution (Thermo Fisher Scientific, Waltham, MA, USA), then stored at −20 °C in the laboratory in order to inhibit RNase enzyme activity until RNA extraction.

RNA extraction and cDNA synthesis

Total RNA was isolated from each samples using TRIzol® Reagent (Thermo Fisher Scientific, Waltham, MA, USA) following the manufacturer’s instructions. The extracted RNA was eluted in 15–30 µl RNase-free water, depending on the pellet size. Yield of RNA was quantified by NanoDrop 1000 Spectrophotometer (Thermo Fisher Scientific, Waltham, MA, USA). To eliminate genomic DNA, samples were treated with RQ1 RNase-Free DNase (Promega, Madison, WI, USA) just before the reverse transcription. First strand cDNA was synthesized from 1 µg DNA-free RNA using the High-Capacity cDNA Reverse Transcription Kit (Applied Biosystems, Foster City, CA, USA).

Reference gene selection and primer design

Using a draft genome of Lethrus apterus (unpublished data) eleven reference genes, which were already described as stable reference genes in other arthropods, were selected (Table 1). We manually designed primers (Table 2) using the web-based Sequence Manipulation Suite (Stothard, 2000) and Multiple Primer Analyzer (Thermo Fisher Scientific, Waltham, MA, USA) in order to avoid the forming of possible secondary structures of the primers. To check the specificity of primer pairs and to determine optimal annealing temperature, PCR reactions were performed in 10 µl volumes containing the following components: 10x buffer, 2 mM MgCl2, 0.2 mM dNTP, 0.02 U/µL Taq DNA polymerase enzymes (DreamTaq Green, Thermo Fisher Scientific, Waltham, MA, USA), 0.2 µM forward and 0.2 µM reverse primer and 0.1 µg cDNA. PCR conditions were optimized by determining the optimal annealing temperature using temperature gradient ranging from 54 °C to 62 °C for primer binding. In this study, we used ABI Veriti® 96-Well Thermal Cycler (Applied Biosystems, Foster City, CA, USA). Cycling conditions consisted of a denaturing step at 95 °C for 2 min followed by 40 cycles at 95 °C for 30 sec, at a temperature gradient (54 °C, 56 °C, 58 °C, 60 °C or 62 °C) for 30 s and at 72 °C for 90 s, and finally at 72 °C for 10 min. PCR amplicons were run on 1% agarose gel stained with GelRed™ (Biotium, Fremont, CA, USA).

Table 1 The list of the candidate housekeeping genes with their biological functions.

Gene	Symbol used	Function	Reference	
glyceraldehyde 3-phosphate dehydrogenase	GAPDH	glycolytic enzyme	Liang et al. (2014)	
tubulin alpha-1 chain	TUB1a	cytoskeletal structural protein	Liang et al. (2014)	
elongation factor 1-alpha	EF1a	protein synthesis	Liang et al. (2014)	
elongation factor 2	EF2	protein synthesis	Zhu et al. (2014)	
ADP-ribosylation factor-like protein 1	ARF1	GTP-binding protein	Shi et al. (2013)	
ADP-ribosylation factor 4	ARF4	GTP-binding protein	Shi et al. (2013)	
ribosomal protein S8	RPS8	structural constituent of ribosome	Yang et al. (2015)	
ribosomal protein L4	RP4	structural constituent of ribosome	Shi et al. (2013)	
ribosomal protein L7A	L7A	structural constituent of ribosome	Zhu et al. (2014)	
ribosomal protein L10	L10	structural constituent of ribosome	Zhu et al. (2014)	
ribosomal protein L18	RP18	structural constituent of ribosome	Shi et al. (2013)	

Table 2 The primers used to measure gene expression levels for the candidate reference genes by RT-qPCR.

Gene	GenBank accession number	Primer sequence (5′to 3′)a	Amplicon length (bp)	Tm (°C)b	E (%)c	R2d	
GAPDH	KY786279	F: GCCATTCCAGTAAGTTTTCCATTGAG	157	85.0	100.75	0.91	
		R: GCTGTTACTGCTACACAAAAGAC					
TUB1a	KY786273	F: CAGACTGCACGTTGGACTTTAGC	172	83.6	100.04	0.96	
		R: TACAGAGGAGATGTTGTCCCCAAG					
EF1a	KY786281	F: AAACCTTTGCGTCTTCCACTACAGG	184	81.7	99.83	0.94	
		R: CTTCAGTTGTAAGACCAACAGGTG					
EF2	KY786280	F: GATGAGAAATCCACATGTCCAG	244	82.0	102.00	0.86	
		R: CGACTCCCTAGTATCAAAGG					
ARF1	KY786283	F: GTATGACAGTAGCTGAAGTTC	141	81.4	112.70	0.84	
		R: CTGTTTTGTAAAGCATTGGC					
ARF4	KY786282	F: TAGTACGGACGGTCAAGTC	197	89.1	105.91	0.81	
		R: GTAGACCGTCACCTGTTATGGC					
RPS8	KY786274	F: CATTATGTACGTACGAGAGGAGGCAACG	200	84.0	99.96	0.91	
		R: TCTAAAGGGAGTAGCGTCGATAACG					
RP4	KY786275	F: TAATGGACCACGACGCTGTATGC	248	84.5	100.33	0.92	
		R: CGTACCAGCTTTAGTAATGAGCAAGG					
L7A	KY786277	F: TAGCGACTCAACTGTTCAAGG	224	84.8	99.54	0.95	
		R: CCTCAATTGGATCGACGTCATGTG					
L10	KY786278	F: CGTAGAGCCTCGATAACTTGG	210	84.7	99.33	0.94	
		R: TCATGTGCTGGAGCTGATAGG					
RP18	KY786276	F: TTGTAACCACATGAACGCCTACG	186	85.2	99.75	0.96	
		R: AGTTAGCTTTACGTTCACCTACTGG					
Notes.

a F, forward primer; R, reverse primer.

b melting temperature.

c real-time qPCR efficiency (calculated by the standard curve method).

d regression coefficient (calculated from the regression line of the standard curve).

Real-time quantitative PCR

RT-qPCR was performed on a QuantStudio 12K Flex Real-Time PCR System (Applied Biosystems, Foster City, CA, USA) using SYBR® Green PCR Master Mix (Applied Biosystems, Foster City, CA, USA) and ROX Passive Reference Dye (Affymetrix, Santa Clara, CA, USA). Amplifications were carried out under the following conditions: initial denaturation at 95 °C for 10 min followed by 40 cycles of 10 sec at 95 °C and for 1 min at the optimal annealing temperature. This was followed by a melting curve analysis in which the temperature raised from 65 °C to 95 °C in sequential steps of 0.05 °C for 1 s. Three technical replicates were performed for each biological sample, and the average cycle threshold (Ct) values of triplicates were calculated. Furthermore, no-template control was done in order to check whether primer-dimers or contamination with amplified PCR product were detectable. Five 5-fold serial dilution was made from cDNA samples to create a standard curve, and the amplification efficiency was determined for each candidate gene. The efficiency (E) values were calculated according to the equation: E = (10(−1∕slope) − 1) × 100, where slope is the slope of the standard curve (Radonić et al., 2004).

Statistical analysis of raw Ct values

In order to examine the differences between sample groups, random intercept mixed-effects models were used with sample id as a random factor for each gene. Significance of fixed terms was investigated by likelihood ratio tests. For likelihood ratio tests models were fitted using Maximum Likelihood estimation. The analyses were carried out using “lme4” (Bates et al., 2014) and “car” (Fox & Weisberg, 2011) packages in the R statistical environment version 3.3.2 (R Core Team, 2016).

Determination of reference gene expression stability

In order to determine the expression stability of the selected reference genes, we used the following methods: geNorm (Vandesompele et al., 2002), NormFinder (Andersen, Jensen & Ørntoft, 2004), BestKeeper (Pfaffl et al., 2004), delta Ct method (Silver et al., 2006) and RefFinder (Xie et al., 2012). For the analyses with the geNorm and NormFinder procedures, the average Ct values were transformed to relative quantities by dividing sample values by the lowest average Ct value. For calculations by BestKeeper, delta Ct method and RefFinder, the untransformed average Ct values were used. All calculations, except the ones done by the web-based RefFinder, were carried out in R with “NormqPCR” package (Perkins et al., 2012).

geNorm calculates the expression stability value M by assessing the mean pairwise expression ratio for each candidate gene against all the other candidates (Vandesompele et al., 2002). The basic assumption of this method is that the expression ratio between two reference genes is identical across the samples. The lower the M value the more stable the expression of the candidate reference gene. Stepwise exclusion of the genes with the highest M value results in the selection of the two most stably expressed reference genes in the tested samples both sharing the same M value. Vandesompele et al. (2002) also suggest not to accept candidate genes as stably expressed reference genes with M value higher than 1.5. Moreover, the procedure determines the normalisation factor by taking the geometric mean of the expression levels from the most stable genes and then additively recalculating with each of the next most stable gene. The pairwise variation, Vn∕n+1 between two sequential normalisation factors is then calculated in order to determine the effect of each newly added gene to the normalisation factor. The optimum number of genes is the lowest number of genes with Vn∕n+1 less than 0.15 (Vandesompele et al., 2002).

NormFinder determines the stability of the candidate reference genes by measuring the intra- and intergroup variation between user specified groups (e.g., male and female groups or treated and control groups) first. Stability values for each candidate gene are then calculated by adding the two sources of variation. The lowest stability value means the most stable expression (Andersen, Jensen & Ørntoft, 2004).

BestKeeper calculates, for each candidate reference gene across the samples, the geometric mean, the arithmetic mean, the minimal and the maximal Ct values, in addition to the average absolute deviation from the arithmetic mean. Genes with the lowest average absolute deviation can be considered as stably expressed reference genes. BestKeeper Index is calculated as the geometric mean of the Ct values of the candidate reference genes. Inter-gene relations are estimated by performing pairwise correlation analyses of all possible reference gene pairs. Furthermore, correlation between the expression level of each candidate gene and the BestKeeper Index is calculated, describing the relation between the index and the contributing genes by the Pearson correlation coefficient, coefficient of determination and the corresponding p-value (Pfaffl et al., 2004).

The delta Ct method compares relative expression of pairs of candidate genes within each sample in order to identify the stably expressed housekeeping genes. If the ΔCt value of the two genes fluctuates when analysed in different samples, it means that one or both genes are variably expressed. If the ΔCt value remains constant, both genes are stably expressed among the samples (Silver et al., 2006).

Each procedure mentioned above uses different algorithms to calculate an expression stability value which represents the suitability of the candidate genes as reference genes, therefore the ranking of the examined genes according to the methods may vary. The web-based tool RefFinder (Xie, 2012) was used in order to combine our results and rank the candidate genes. This user-friendly program integrates the four methods mentioned above. Using the ranking from each program, it assigns an appropriate weight to an individual gene and calculates the geometric mean of their weights for the overall ranking. The lowest rank indicates the most stably expressed gene (Xie et al., 2012).

For each analysis, except for NormFinder, seven sample groups were used: all samples irrespective of body part or sex; head samples irrespective of sex; male head samples; female head samples; thorax samples irrespective of sex; male thorax samples; female thorax samples. The calculation by NormFinder requires subgroup specification, therefore, body regions were set as subgroups for the analysis of all samples. In order to investigate the effect of sexes, male and female subgroups were specified for the analysis of head and thorax sample groups separately. In this way, three sample groups, each divided into two subgroups, were analysed by NormFinder.

Results

Transcriptional profiling of candidate reference genes

Before the evaluation of expression stability of the eleven candidate genes, specificity of each primer pair was checked on 1% agarose gel which showed single products with the expected sizes. Moreover, gene-specific amplification was confirmed by single melting curve peaks. These results indicate that no primer-dimers or nonspecific amplification products were formed. Additionally, no fluorescent signals were detected in the negative control during the RT-qPCR. Each amplicons were sequenced and annotated to the sequences from which the primer design was based in order to check that the correct genes were amplified. The sequences are available in File S1. The efficiency of the eleven candidate genes ranged from 99.33 to 112.70%. The efficiency values and other basic information of the RT-qPCR required based on the guideline of Bustin et al. (2009) are included in Table 2.

Raw Ct values ranged from 11.66 (TUB1a) to 30.12 (ARF4) (Fig. 1). The mean and standard deviation (SD) of the Ct values across all samples were calculated for each gene (Table 3). Since the mean Ct values ranged between 15 and 30 for all the candidate reference genes, all of them were analysed further (Kozera & Rapacz, 2013). ARF1 had the least variable expression level with the lowest SD value (SD = 1.85), while ARF4 had the most variable expression level (SD = 3.04). Low average Ct values indicate high expression level in TUB1a and EF2 (Ctmean = 15.11), on the other hand, high Ct values of ARF1 (Ctmean = 20.77) indicated low expression.

Figure 1 Expression profiles of the 11 candidate reference genes.

Table 3 Mean, standard deviation and coefficient of variation for the Ct values of 11 candidate reference genes, calculated across all samples.

Genes	Mean	SD	CV	
GAPDH	15.27	2.62	0.17157826	
TUB1a	15.11	2.64	0.17471873	
EF1a	15.44	2.34	0.1515544	
EF2	15.11	2.05	0.13567174	
ARF1	20.77	1.85	0.08907078	
ARF4	20.22	3.04	0.15034619	
RPS8	16.38	2.12	0.12942613	
RP4	15.52	1.94	0.125	
L7A	17.27	2.1	0.12159815	
L10	16.15	1.97	0.12198142	
RP18	16.02	1.93	0.12047441	

Based on the likelihood ratio tests, sex had no significant effect on the expression level of the candidate genes. However, significant effect of body region was found in case of six genes: GAPDH, EF1a, ARF1, ARF4, RP4 and L10. The interaction of sex and bodypart had no significant effect on the expression level of the candidates, except for RPS8 (Table 4).

Table 4 Results of likelihood ratio tests on the effects of body region, sex and their interaction on the expression levels of the eleven candidate reference genes.

Gene	Sex	Bodypart	Sex*Bodypart	
	χ2	P	χ2	P	χ2	P	
GAPDH	0.019	0.889	5.668	0.017	0.965	0.326	
TUB1a	0.642	0.423	1.960	0.162	1.995	0.158	
EF1a	0.002	0.968	4.096	0.043	0.857	0.355	
EF2	1.216	0.270	2.271	0.132	2.030	0.154	
ARF1	0.476	0.490	4.147	0.042	2.459	0.117	
ARF4	1.417	0.234	4.885	0.027	1.289	0.256	
RPS8	0.586	0.444	2.307	0.129	4.007	0.045	
RP4	0.356	0.551	4.272	0.039	2.384	0.127	
L7A	0.556	0.456	3.087	0.079	2.901	0.089	
L10	0.246	0.620	8.509	0.004	3.319	0.068	
RP18	1.028	0.311	2.336	0.126	2.896	0.089	
Notes.

Significant effects are highlighted in bold.

Expression stability of candidate reference genes

Based on geNorm analysis for all samples, eight candidate genes had an M value below the threshold of 1.5 (Table S1). The results show that the lowest M value was 0.390 for RPS8 and L7A. Among the head samples irrespective of sex, all of the tested genes except L10 had an M value below 1.5, and RPS8 and RP18 were co-ranked as the most stable genes from the candidates (M = 0.304). Furthermore, the same two genes had the lowest M value considering male and female head samples separately (M = 0.264 for females and M = 0.346 for males). In case of the thorax samples irrespective of sex, eight genes had an M value below the threshold. RPS8 and L7A were the most stable candidate gene pair with an M value of 0.358. In thorax samples collected from females, RPS8 and RP18 were the most stable genes as well with an M value of 0.222. However, in thorax samples of males, RPS8 and L7A were ranked as the best reference gene pair (M = 0.288).

According to NormFinder, L7A was the most stable gene when calculating with all samples divided into groups of head and thorax samples (Table S2). The second and third genes were RP4 and RPS8, indicating that these are also worth considering as reference genes. In the case of specifying males and females as subgroups within head and thorax samples, L7A was found again to be the most stably expressed gene among the candidate ones. In both head and thorax samples, L7A was followed by similar ranking order: EF2, RP4, RP18 and RPS8 as second, third, fourth and fifth genes, respectively.

Based on BestKeeper, across all samples ARF1 had the lowest mean absolute deviation (MAD) value; however, L7A had the highest correlation r value (Table S3). In the group of head samples irrespective of sex, ARF1 had the lowest MAD value, while among the thorax samples irrespective of sex, L10 was the most stable according to the MAD value. This was surprising as the other programs ranked this gene consistently as one of the least stable genes. On the other hand, in both head and thorax samples, L7A had the highest r value. In head samples of males RPS8 (MAD = 0.827), and of females ARF1 (MAD = 1.122) were the most stable candidate genes. In both male and female head samples, L7A had the highest correlation r value. Considering male thorax samples L10 had the lowest MAD value (MAD = 1.576), while in female thorax samples EF2 was ranked as the most stable with MAD value 1.236. L7A had the highest r value in female thorax samples, however, in male thorax samples EF2 had the highest correlation r value.

According to the delta Ct method, L7A was the most stable gene among the candidates overall with the stability value always below 1.0 (Table S4).

Finally, the candidate genes were evaluated by RefFinder to combine the results of individual methods (Table 5). Using all samples irrespective of body region and sex, and separately the head and thorax samples irrespective of sex, L7A was ranked first, as the most stably expressed gene among the candidate reference genes. In head samples, RP18 was co-ranked with L7A as the most stable reference genes. In thorax samples, RP4 was ranked on the second place. In female head and thorax samples, RP18 was the most stably expressed gene of the candidates. Considering head samples of males, RPS8 was ranked on the first place, while in thorax samples of males, L7A was ranked as the best reference gene. L7A was ranked on the second place in all subgroups, with the exception of male thorax samples, where EF2 was the second best reference gene according to RefFinder.

Table 5 Stability ranking of the eleven candidate reference genes in the different sample groups as calculated by RefFinder.

Rank	All samples	Head samples	Thorax samples	
		All head samples	Male head samples	Female head samples	All thorax samples	Male thorax samples	Female thorax samples	
1	L7A	L7A	RPS8	RP18	L7A	L7A	RP18	
2	RP18	RP18	L7A	L7A	RP4	EF2	L7A	
3	RPS8	RPS8	RP18	RPS8	RP18	RPS8	EF2	
4	EF2	ARF1	EF2	EF2	RPS8	RP4	RPS8	
5	ARF1	RP4	TUB1a	ARF1	EF2	L10	RP4	
6	RP4	EF2	RP4	RP4	L10	EF1a	L10	
7	EF1a	GAPDH	ARF1	EF1a	ARF1	RP18	ARF1	
8	TUB1a	TUB1a	GAPDH	GAPDH	EF1a	ARF1	EF1a	
9	L10	EF1a	L10	ARF4	TUB1a	ARF4	TUB1a	
10	GAPDH	L10	ARF4	L10	ARF4	TUB1a	GAPDH	
11	ARF4	ARF4	EF1a	TUB1a	GAPDH	GAPDH	ARF4	

Optimal number of reference genes

To determine the minimal number of genes necessary for normalisation, the V-value was computed by geNorm. The results demonstrated that across all samples V2∕3 was the first V-value lower than the cut-off value of 0.15 (Fig. 2). Considering separately the head and thorax samples, V2∕3 was again lower than 0.15. Separate analyses of female and male samples within head and thorax groups showed that V2∕3 was also the first value below the threshold in all cases (results not shown). Therefore, two stably expressed reference genes are sufficient for normalisation in any case of sample classification.

Figure 2 Pairwise variation analyses by geNorm to determine the optimal number of reference genes for accurate normalization.

Pairwise variation for all samples together, as well as separately for head and thorax samples. The lowest number of genes with Vn∕n+1 less than 0.15 means the optimum number of genes.

Discussion

RT-qPCR is a widely used method for measuring gene expression levels due to its relatively low cost, high accuracy and sensitivity. A critical step of this method is data normalisation which requires careful selection of reference genes for the given experimental or environmental conditions. With these stably expressed genes, technical errors and variance resulting from the method can be moderated (Udvardi, Czechowski & Scheible, 2008). Several studies have examined the stability of reference genes in various insect species in the past decade and these studies suggest that no universally stable reference gene can be found that is applicable for all species, tissue types and experimental conditions. Hence, it is necessary to identify the most suitable reference genes for the specific circumstances in a given study for a given species (Zhu et al., 2014).

In the present study, variation in expression levels of eleven housekeeping genes were evaluated across a span of 1.5 months covering most of the breeding period of the biparental beetle Lethrus apterus. To date, no study investigated the possible reference genes either in this species, or in the family of Geotrupidae. We analyzed the expression stability of the candidate reference genes by four frequently used programs: geNorm, NormFinder, BestKeeper and comparative delta Ct method. The outcomes of these programs can vary because of the differences in the algorithms. Therefore, the combined use of them ensures more reliable results. For this purpose, RefFinder, a freely available web-based tool was used to calculate a comprehensive ranking value for each candidate gene.

According to the comprehensive ranking by RefFinder, the most stably expressed reference gene was L7A across all samples, irrespective of body region and sex. Based on the results of geNorm analysis, two reference genes are sufficient for normalisation in gene expression analysis in Lethrus apterus during the breeding period. For accurate normalisation, we recommend the use of L7A and RP18 in head samples irrespective of sex. When considering the sexes separately, RPS8 and L7A should be used for head samples of males, and RP18 and L7A for females. In thorax samples irrespective of sex, L7A and RP4 are the best reference genes. In case of thorax samples, L7A and EF2 are recommended for normalisation in males, RP18 and L7A in females.

Consistent with our results, ribosomal proteins are reported to be the best reference genes in many insect species. In a study by Zhu et al. (2014), ribosomal protein L7A was ranked as one of the best reference genes in Spodoptera exigua in different tissues, specific larval physiological stages and male individuals. Studies of other coleopterans gave similar results: RP4 and RP18 were the best reference genes in Leptinotarsa decemlineata (Shi et al., 2013), RPS3 (ribosomal protein S3), RPL13a (ribosomal protein 13a) and RPS18 (ribosomal protein S18) were suitable reference genes for Tribolium castaneum (Lord et al., 2010; Sang et al., 2015), and RPL22e (ribosomal protein 22e) was one of the best reference genes in Mylabris cichorii both in males and females (Wang et al., 2014). In other species, e.g., in Drosophila melanogaste r Rpl32 (ribosomal protein L32) was a suitable reference gene in individuals on different diets (Ponton et al., 2011), and in Aphis craccivora, RPS8, RPL14 (ribosomal protein L14), and RPL11 (ribosomal protein L11) were the three most stable housekeeping genes across different developmental stages and temperature conditions (Yang et al., 2015).

Interestingly, two frequently used reference genes, GAPDH and TUB1a were ranked as less stable genes in this study, beside ARF4, with stability values above the threshold values of all the programs used. L10 was also found to be an unstable candidate gene in all but the geNorm analysis. These results correspond with the findings of Thellin et al. (1999), i.e., housekeeping genes should be evaluated as reference genes across the given experimental conditions in the given species. Based on our results, we recommend to avoid the use of these last four genes for normalisation in studies investigating gene expression patterns during the reproductive period in this species.

Conclusion

By evaluating the stability of eleven candidate housekeeping genes in samples collected during the breeding period of free-living Lethrus apterus, we conclude that two of them provide sufficient reference for normalising target gene expression. In head samples, these two genes appear to be L7A and RP18, whereas in thorax samples L7A and RP4 should be used. In both thorax and head samples of females, RP18 and L7A are the best choices for normalisation. Based on our results, in head samples of males, RPS8 and L7A, while in thorax samples of males, L7A and EF2 are recommended to use. These results provide reliable reference genes that are suitable normalizers for further RT-qPCR investigations on the hormonal regulation in Lethrus apterus.

Supplemental Information

Table S1 Expression stability ( M) values of the eleven candidate reference genes in the different sample groups as calculated by geNorm

Lower M value indicates more stable expression. The most stable gene pair in each sample group is highlighted in bold.

Click here for additional data file.

Table S2 Stability ranking of the candidate reference genes calculated by NormFinder

As NormFinder requires setting subgroups, the specified subgroups were as follows: head and thorax samples within all samples; male and female samples within head samples; male and female samples within thorax samples, respectively. Because of this requirement we do not have the same sample groups as analysed with the other procedures.

Click here for additional data file.

Table S3 Mean absolute deviation (MAD) and Pearson correlation coefficient (r) of the candidate reference genes in the different samples groups as calculated by BestKeeper

Lower MAD value and higher r value means more stable expression. The most stably expressed genes are highlighted in bold.

Click here for additional data file.

Table S4 Mean standard deviation (SDavg) calculated by delta Ct method

Lower SDavg value indicates more stable expression. The best candidate genes are highlighted in bold.

Click here for additional data file.

File S1 Housekeeping gene sequences as uploaded to GenBank

Click here for additional data file.

File S2 Sequences of the products amplified by real-time qPCR

Click here for additional data file.

We are grateful to Tamás Varga for allowing us to conduct fieldwork on his property and Adrien Fónagy for helping us in the development of the autopsy.

Additional Information and Declarations

Competing Interests

Author Contributions

Field Study Permissions

DNA Deposition

The authors declare there are no competing interests.

Nikoletta A. Nagy conceived and designed the experiments, performed the experiments, analyzed the data, wrote the paper, prepared figures and/or tables, reviewed drafts of the paper.

Zoltán Németh and András Kosztolányi conceived and designed the experiments, analyzed the data, wrote the paper, reviewed drafts of the paper.

Edit Juhász, Szilárd Póliska and Rita Rácz conceived and designed the experiments, performed the experiments, wrote the paper, reviewed drafts of the paper.

Zoltán Barta conceived and designed the experiments, analyzed the data, contributed reagents/materials/analysis tools, wrote the paper, reviewed drafts of the paper.

The following information was supplied relating to field study approvals (i.e., approving body and any reference numbers):

Sample collection was approved by the Northern Hungarian Inspectorate for Environment Protection and Nature Conservation.

The following information was supplied regarding the deposition of DNA sequences:

The housekeeping gene sequences described here are available via GenBank accession numbers KY786273 to KY786283.

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
