# Peer review of "Evaluation of potential reference genes for real-time qPCR analysis in a biparental beetle, Lethrus apterus (Coleoptera: Geotrupidae)"

_PeerJ, doi:10.7717/peerj.4047_

## Round 0.1 · original submission · Major Revisions

I think that the manuscript is well-written and technically sound, but some important revisions are needed prior to publication.

In particular, I strongly encourage the authors to provide some significant details on the experimental design and the results, as follow:
- describe carefully the different subset of analysed samples;
- provide qPCR parameters for primer pairs;
- provide RefFinder results;
- clarify data shown in the tables (especially 4, 5, 6).
(for other specific comments, see also the reviewers’ suggestions).

The authors should also provide a statistical analysis for data shown in fig 1.

A sequence of PCR products is also needed in order to confirm that the right gene has been amplified.

As far as the comment of reviewer 2 on the importance to test the candidate genes in different stages of development and other variable situation, in my opinion it is not necessary because in the paper it has been clearly stated that the reference genes can be used for studies of gene expression in specimens during the breeding period (i.e. during adult stage), in the specific body regions of the thorax and head.

For all the reasons specified above, my decision is major revisions.

Reviewer 1 ·

Basic reporting

The ms is clear, well written in professional English and conforms to professional standards of expression.
Literature references basically cover sufficiently the field context. I would suggest to add the following paper and to accomplish the checklist provided in it.
The MIQE Guidelines: Minimum Information for Publication of Quantitative Real-Time PCR Experiments, by Stephen A. Bustin et al., Clinical Chemistry Apr 2009, 55 (4) 611-622; DOI: 10.1373/clinchem.2008.112797.
The ms includes sufficient introduction and background and demonstrates how the work fits into the broader field of knowledge.
Figures are relevant to the content of the article and appropriately described and labeled. Resolution of Figure 1 is probably too poor. It is suitable for review, but it would be necessary improve the quality in final version. A comment to figure 1 is missing in result sections. Figure 1 indeed shows that generally thorax samples showed higher Cts than head, especially in males and that generally there were no strong differences between male and female samples. A statistical analysis of these differences (ANOVA on different samples for each genes) could improve this statement and Figure 1 as well.
A different table organization is needed to improve ms readability. Tables 1 and 2 could be merged, and Authors should add qPCR parameters for different primer pairs that are now missing (Efficiency, R2, melting temperature). Table 5 and 6 should be provided as supplementary data, since are not crucial. On the other hand, data about different ranking obtained by reffinder on different tissue and different sex are missing. Authors should include new table with reffinder results for different subset of samples analysed, especially to facilitate reading of lines 251-258 in results and 296-300 in discussion. These paragraphs describe key findings of ms, but they need to be rephrased.
The ms is ‘self-contained’ and represents an appropriate ‘unit of publication’.

Experimental design

Research question is well defined, relevant and meaningful, as author stated in discussion lines 284-287.
The investigation was rigorous and performed to a high technical standard.

Methods are generally well described. Nevertheless, the different subset of samples analysed should be clearly stated. In other words, if I understood well, authors run different gene stability analysis to establish reference genes on different sample subsets: i) all samples, irrespective of tissue and sex; ii) all females, irrespective of tissue; iii) all males, irrespective of tissue; iv) all thorax samples, irrespective of sex; v) all head samples, irrespective of sex; vi) female thorax samples; vii) male thorax samples; viii) female head samples; ix) male head samples. Authors should better explain this experimental plan of analyses and including a new table at least including reffinder results for all these analyses.

Validity of the findings

Presented results are statistically sound, but some details on different subset of analysis are missing, as previously indicated.
qPCR Efficiency must be calculated for all primer pairs. Indeed in relative quantity calculation it should be included to obtain more accurate data.
Conclusion are well stated, linked to original research question and limited to supporting results.

Additional comments

Detailed comments not included in above suggestions are:
-In the abstract is missing the fact that L7A and RP18 are also the most stable genes considering all samples irrespective of tissues and sex.
- “Lethrus apterus”, “reference gene” and “RT-qPCR” keywords are redundant since already present in the title.
-Lines 58-42: please rephrase.
- Line 84: Change it, since, reference genes must be evaluated in different specific sample types and treatments. The concept of “universally applicable reference genes” is not reasonable.
- Line 89: change in “the expression level of hormone regulating genes involved in parental care”
- Line 141-142: please add details about negative controls of no-retrotranscribed samples to exclude genomic DNA contamination.
-Line 148 149. Change with the “lowest Ct value” instead of the “highest”. (the lowest Ct value corresponds to the highest cDNA starting quantity)
-Caption to Table 5: rephrase the subset of samples analysed.

Reviewer 2 ·

Basic reporting

No comment

Experimental design

1. In line 116, the author quoted “Using a draft genome of Lethrus apterus (Rácz et al., 2015)”. Looking at the paper, its not a genome paper, it’s a paper identifying microsatellite markers. Can the author clarify this? Is there a genome of Lethrus apterus?
2. The amplicon length of some of the genes are quite big, up to 426bp for a qPCR experiment. Its usually recommended to keep amplicon size small and if possible about the same length across all test genes. Is there any particular reason why some of these genes have big amplicons?
3. The authors mention about sample collection in line 94-96. Was the sample pooled? If so, which samples were pooled?
4. The number of variables are quite limited, only head and thorax of male and female collection. Although it is understandable that the sample is an environmental sample, but in this type of experiment, it is important to test the candidate genes in different variability (e.g. different stages of development) to measure its stability.

Validity of the findings

5. For the gene finder and BestKeeper analysis in Table 4 and 6. The results shown are for which samples? As there are 4 samples chosen by the authors (head and thorax of male and female), why is there only 1 set of data? Can the authors comment to make this clearer.
6. Same goes to NormFinder data in Table 5, why is there only Head and Thorax but there is not male and female data?
7. Did the authors sequence the PCR products to see if the correct genes were amplified?
8. Could the author explain why the housekeeping gene was not tested/validated using another gene?

Additional comments

The work presented in this manuscript describe the use of potential reference genes for the use of qPCR based studies in Lethrus apterus. The authors choose 11 candidate genes to test its suitability to become the housekeeping genes.
The data presented appeared to be technically sound for the most part. Most papers are using geNorm, NormFinder, BestKeeper and RefFinder to evaluate the suitability of these candidate genes.

---

## Round 0.2 · accepted · Accept

In my opinion the article has been substantially improved. The authors provided all the missing details and data as requested by the reviewers, and the manuscript is now ready to be published.